# An Inducible System for Silencing Establishment Reveals a Stepwise Mechanism in Which Anchoring at the Nuclear Periphery Precedes Heterochromatin Formation

**DOI:** 10.3390/cells10112810

**Published:** 2021-10-20

**Authors:** Isabelle Loïodice, Mickael Garnier, Ivaylo Nikolov, Angela Taddei

**Affiliations:** 1Nuclear Dynamics Unit, CNRS, Institut Curie, PSL University, Sorbonne Université, 75005 Paris, France; isabelle.loiodice@curie.fr (I.L.); mickael.garnier@curie.fr (M.G.); ivaylo.l.nikolov@gmail.com (I.N.); 2Cogitamus Laboratory, F-75005 Paris, France

**Keywords:** chromatin, heterochromatin, nuclear organization, gene regulation, gene silencing, FROS

## Abstract

In eukaryotic cells, silent chromatin is mainly found at the nuclear periphery forming subnuclear compartments that favor silencing establishment. Here, we set up an inducible system to monitor silencing establishment at an ectopic locus in relation with its subnuclear localization in budding yeast. We previously showed that introducing LacI bound *lacO* arrays in proximity to gene flanked by *HML* silencers favors the recruitment of the yeast silencing complex SIR at this locus, leading to its silencing and anchoring at the nuclear periphery. Using an inducible version of this system, we show that silencing establishment is a stepwise process occurring over several cell cycles, with the progressive recruitment of the SIR complex. In contrast, we observed a rapid, SIR-independent perinuclear anchoring, induced by the high amount of LacI binding at the *lacO* array leading to nucleosome eviction at this array and to the phosphorylation of H2A in the neighboring nucleosomes by Mec1 kinase. While the initial phosphorylation of H2A (H2A-P) and perinuclear anchoring are independent of the SIR complex, its latter recruitment stabilizes H2A-P and reinforces the perinuclear anchoring. Finally, we showed that Sir3 spreading stabilizes nucleosomes and limits the access of specific DNA-binding protein to DNA.

## 1. Introduction

In eukaryotic cells, discrete regions of the genome assume heritable chromatin structures that silence genes located in these regions. In most organisms, and most cell types, silent chromatin or heterochromatin is enriched at the nuclear periphery forming subnuclear compartments where general repressors of transcription concentrate, thus favoring silencing establishment at the nuclear periphery [1,2].

In budding yeast, silent chromatin is mainly found in subtelomeric regions, at the two cryptic mating type loci (*HML* and *HMR*) and at the ribosomal DNA locus. In cycling cells, telomeres and *HM* loci are preferentially located at the nuclear envelope, forming repressive domains that sequester the yeast silencing factors (SIRs) [3] in a manner comparable to heterochromatic chromocenters, which sequester HP1 in metazoans [1,4].

Similarly to HP1, the yeast silencing complex SIR2/3/4 is able to spread along the chromatin and to repress gene transcription owing to the histone deacetylase activity of Sir2 that creates a binding surface for Sir3 [3]. This spreading is limited by the amount of Sir3 and by the methylation of H3K79, which prevents Sir3 binding to nucleosomes [5].

At telomeres, silencing is nucleated at the subtelomeric TG1-3 repeats by the binding of the transcription factor Rap1 that contains binding sites for the silencing factors Sir3 and Sir4 at its C-terminal end. At the cryptic mating type loci, recruitment of the SIR complex requires the presence of cis acting elements called silencers that are formed by combinations of binding sites for DNA binding proteins including Rap1, Orc1, and Abf1, which like Rap1 have other functions in the cell. A common feature of these factors is their strong ability to displace nucleosomes [6].

At the rDNA, Sir2 is recruited by Net1 that associates with the RNA-polymerase I and Fob1 that binds the rDNA replication fork barrier as part of the RENT complex [7]. While Sir2, Sir3, and Sir4 are all required for silencing at *HM* loci and subtelomeres, Sir2 only is required for silencing at the rDNA, although Sir3 is found associated with the rDNA by ChIP and live microscopy [8,9].

In addition to these canonical sites, Sir3 is also found by ChIP at few sites located on chromosome arms [5,10,11,12]. In contrast to the situation at *HM* loci and subtelomeres, Sir3 does not spread from these recruitment sites owing to the presence of H3K79me3 nucleosomes on the flanking regions [5]. What triggers the recruitment of Sir3 at these sites is not known. Even at canonical silenced loci, it is not clear how the combination of binding sites for factors with independent roles in the cell can create a silencer able to nucleate gene silencing. It is generally admitted that the property of silencers emerges from the close juxtaposition of factors that interact directly or indirectly with one or more Sir proteins: a sufficiently high local concentration of Sir proteins to sustain silencing being achieved only when multiple binding sites are combined. However, there is no clear evidence for Abf1 interaction with any of the SIR proteins, suggesting that other mechanisms must be involved.

An important feature of heterochromatin in budding yeast is the limiting amount of the silencing factors Sir2 and Sir3, leading to an intranuclear competition between their target sites [13,14,15]. Another consequence of the limiting amount of SIR protein is that the efficiency of *HM* silencers is sensitive to the distance to telomeres, which are located in compartments of the nucleoplasm enriched in silencing factors [16]. Accordingly, reporter genes flanked by functional silencers are weakly repressed by the SIR complex when integrated away from a telomere [17,18,19,20,21].

We previously showed that introducing tight protein–DNA complexes such as LacI bound *lacO* arrays in proximity to a reporter gene favor the recruitment of the SIR complex in this context [17], acting as a proto-silencer promoting gene silencing and change in subnuclear localization.

As mentioned above, silent chromatin is preferentially associated with the nuclear periphery due to different anchoring pathways, one of them involving Sir4 association with a network of docking sites on the inner nuclear membrane [16]. Reciprocally, the artificial tethering of a gene to the nuclear periphery favors SIR-mediated silencing [22] due to the high local concentration of SIR protein in this area of the nucleus [21]. Peripheral anchoring and SIR spreading thus reinforce each other once heterochromatin is established. However, their respective contribution for the establishment of silent chromatin is not known.

Heterochromatin establishment has been mainly studied using different strategies, most of them using conditional or inducible *SIR3* alleles [23,24,25,26,27,28,29,30,31,32], thus affecting all silent loci and possibly changing the stoichiometry of the SIR complex. Here, we developed and characterized an inducible system to study the establishment of silencing at a single ectopic locus, while silencing is already established at canonical loci, without affecting the levels of silencing factors. We used this system to explore the relationship between histone modifications, perinuclear anchoring, and silencing factor recruitment upon silencing establishment.

## 2. Materials and Methods

### 2.1. Media and Growth Conditions

Yeast cells were grown either in rich medium (YP) supplemented with 2% glucose; 3% raffinose or 2% galactose (wt/vol) or in synthetic medium (YNBAS MP Biomedicals, Solon, OH, USA) plates or liquid enriched for 2× dropout CSM-TRP (MP Biomedicals) and supplemented with 2% glucose (*w*/*v*) (referred as CSM-TRP plates or medium). All the strains were grown at 30 °C with shaking at 250 rpm. For galactose induction, YP was supplemented with 0,008% adenine-HCl (SIGMA A8751, Sigma-Aldrich, St. Louis, MO, USA). Typically, (for Figures1–6 but Figure 6F and Appendix A) on day one, cells were thawed on YPD or CSM-TRP plates; on day two, cells were grown 6 h to 7 h in liquid YPAD and diluted at 0.01 OD_600_ nm overnight in YPA 3% raffinose; on day three, cells were diluted at 0,2 OD_600_ nm in YPA 3% raffinose for 2 h then split in two. Galactose (2% final) was added to one half of the culture for galactose induction, the other half was kept in the same medium YPA 3% raffinose as a control, and cells were put back to grow. At the end of day 3, both cultures were diluted at 0.005 OD_600_ nm and grew overnight to obtain the 20 h (or 1200 min) time point.

For Figure 6F and Appendix A, freshly thawed strains on CSM-TRP plates were grown 6 h to 7 h in liquid 2 × CSM-TRP medium, then diluted at 0.01 OD_600_ nm overnight in 2 × CSM-TRP medium. The next day, cells were diluted at 0.2 OD_600_ nm in 2 × CSM-TRP medium and imaged at 1 OD_600_ nm.

### 2.2. Strains

The strains used in this study are listed in Appendix A. They are all derivatives of W303 [33], except for yAT1798, yAT1909, and yAT2142 (S288C, [34]). Gene deletions, insertions of alternative promoters, and gene tagging were performed by PCR-based gene targeting on plasmids from [35,36].

### 2.3. Plasmids

The plasmid encoding the GFP-LacI construct to be integrated at the ADE2 locus (pAT378) was constructed as described below: a part of the *ADE2* ORF (nucleotide144-950) was amplified by PCR from pAT147 (pEADEI from [18]) using primers oAT884 and oAT885 that contain, respectively, in their 5′ extremity a SacI and SacII cutting sites. The PCR product was digested and cloned into pAT123 [17] digested by SacI and SacII yielding pAT376. A second PCR product spanning from nucleotide 1278 of *ADE2* to nucleotide 2635 of *RGA1* was amplified from pAT147 using primers oAT995 and oAT996 that contain, respectively, in their 5′ extremity a SacI and NaeI cutting sites. This PCR product was digested and cloned into pAT376 digested by SacI and NaeI to obtain pAT377. Then, pAT377 was cut by HindIII and religated to obtain the final plasmid pAT378. Another version of this plasmid encoding a LacI protein resistant to galactose (LacI R197K) was obtained by directed mutagenesis on pAT378 using primers oAT1221 and oAT1222, yielding pAT391. pAT390, another version of the GFP-LacIR plasmid that is integrating at the LEU2 locus, was obtained by directed mutagenesis on pAT123 using primers oAT1168 and oAT1169. See sequences of the primers in Appendix A Appendix A.

### 2.4. ChIP and Quantitative PCR Analysis

A total of 20 OD_600_ nm equivalent of cells was fixed in 20 mL with 0.9% formaldehyde for 15 min at 30 °C, quenched with 0.125 M glycine for 5 min, and washed once in cold 1× TBS pH 7.6. Pellets were suspended in 1 mL 1× TBS, centrifuged again, and frozen in liquid nitrogen after removal of the supernatant for −80 °C storage. All the following steps were done at 4 °C unless indicated. Pellets were resuspended in 500 μL of lysis buffer (0.01% SDS, 1.1% Triton X-100, 1.2 mM EDTA pH8, 16.7 mM Tris pH 8, 167 mM NaCl, 0.5% BSA, 0.02 g/L tRNA, and 2.5 μL of protease inhibitor from Sigma-Aldrich P1860, Sigma-Aldrich, St. Louis, MO, USA) and mechanically lysed using a FastPrep instrument (MPBiomedicals) with 0.5 mm zirconium beads (Biospec Products, Bartlesville, OK, USA): intensity 6, 3 cycles of 30 s with 3 min incubation on ice in between cycles. The chromatin was fragmented to a mean size of 500 bp by either sonication using the Bioruptor XL (Diagenode, Liège, Belgium) for 14 min at high power with 30 s on/30 s off, or sonication in the Bioruptor Pico at high intensity for 3 cycles of 30 s on/30 s off. The extracts were cleared by centrifugation for 5 min at 16,000 g. Ten microliters was kept to be used as input. Cleared lysates were incubated overnight with the antibody used for chromatin immunoprecipitation: 1 μL of polyclonal antibody anti-Sir3 (Agro-bio) (Ruault et al. 2011), 2 µL of custom made polyclonal anti-Sir4 (Proteogenix, Schiltigheim, France), 3 µL of polyclonal anti-H2A (Active motif, 39,235), 1 µL of polyclonal anti-Histone H2A (phospho S129) (Abcam, ab15083, Cambridge, UK), and 4 µL of polyclonal anti-acetyl-Histone H4 (Lys16) (Merck-Millipore, 2073125, Burlington, MA, USA). The next day, 50 µL of magnetic beads protein A (NEB) was added to the extracts/antibody mixture and incubated for 4 h at 4 °C on a rotating wheel. Then, the magnetic beads with the immunoprecipitated material were washed sequentially once with lysis buffer, twice with RIPA buffer (0.1% SDS, 10 mM Tris pH 7.6, 1 mM EDTA pH8, 0.1% sodium deoxycholate, and 1% Triton X-100), twice with RIPA buffer supplemented with 300 mM NaCl, twice in LiCl buffer (250 mM LiCl, 0.5% NP-40, 0.5% sodium deoxycholate), once with 1× TE 0.2% Triton X-100, and a final was in 1× TE. The beads were then resuspended in 100 µL of elution buffer and placed in an incubator at 65 °C with gentle agitation to elute the immunoprecipitated material from the beads. In the meantime, the inputs were diluted 1/10 with elution buffer. A reversal cross-linking was performed by heating the samples, inputs and IP, overnight at 65 °C. Proteins were digested with Proteinase K (0.4 mg/mL) in the presence of glycogen, and the remaining DNA was purified on QIAquick PCR purification columns. Finally, samples were treated with 29 μg/mL RNase A for 30 min at 37 °C and used for quantitative PCR.

ChIP quantification was performed by quantitative PCR (qPCR) either on the 7900HT Fast Real-Time PCR (Applied Biosystems, Waltham, MA, USA) or on the QuantStudio 5 Real-Time PCR System (Applied Biosystems, Waltham, MA, USA). Sequences of interest were amplified using the SYBR Green PCR Master Mix (Applied Biosystems, Waltham, MA, USA) and the primers listed in Appendix A on a dilution of immunoprecipitated DNA at 1/40 and 1/80 for the input DNA. PCR reactions were conducted using the following program: an initial denaturation at 95 °C for 10 min followed by 40 cycles (95 °C for 15 s and 60 °C for 30 s). Each real-time PCR reaction was performed in triplicate. The signal from a given region was normalized to the one from the *OGG1* control locus in immunoprecipitated and input DNA samples. Plots represent the mean value obtained for at least three independent experiments; data are shown as mean ± s.e.m.; see Appendix A for the number of experiments.

### 2.5. Quantitative Transcript Analysis

Total RNA was isolated from yeast using the RNeasy kit (Qiagen, Hilden, Germany) followed by a DNase treatment (Qiagen, 79254, Hilden, Germany) to remove contaminating DNA. First strand cDNA was prepared from 0.5 µg of RNA and SuperScript™ III First-Strand Synthesis SuperMix for qRT-PCR (ThermoFisher, 11752050, Waltham, MA, USA). Quantitative PCR was performed on a 1/50 dilution of the reverse transcript mix with a couple of primers designed in *ADE2* and a couple of primers designed in *ACT1* (sequences provided in Appendix A). Values were normalized by *ACT1*, then, to account for the change in expression induced by galactose induction independently of LacI induction, a ratio was made between strains with LacI expression (wt const. yAT2059 and wt ind. yAT2078) and a strain without LacI expression (wt No LacI yAT2001).

### 2.6. Microscopy

The set of images from any given figure panel was acquired using identical acquisition parameters. For all fluorescent images, the axial (z) step is 200 nm, and images shown are a maximum intensity projection. Images were acquired using a wide-field microscopy system driven by MetaMorph software version 7.8 (Molecular Devices, Silicon Valley, CA, USA) based on an inverted microscope (Nikon TE2000, Nikon, Minato City, Tokyo, Japan) equipped with a 100×~/1.4 NA immersion objective, a C-mos camera (ORCA-flash C11440, Hamamatsu, Japan), and either a xenon arc lamp (Sutter Instrument Co. Lambda LS, Novato, CA, USA) or a Spectra X light engine lamp (Lumencor, Inc, Beaverton, OR, USA).

### 2.7. Image Processing and Analysis

Deconvolution of images acquired with the wide-field microscopy system was made using the Meinel algorithm in MetaMorph (eight iterations; sigma = 0.8; frequency 3; MDS Analytical Technologies, Sunnyvale, CA, USA).

All the microscopy image quantifications were done using either a single Fiji macro or MATLAB scripts with an additional expert supervision step to ensure that only relevant data are included in the statistical analysis.

### 2.8. Spot Localization

The spot localizations are extracted using a Fiji macro applying Laplacian of Gaussian filters on pre-processed images followed by an h-maxima transform. Due to the presence of highly variable signals present inside the yeast vacuoles, no reliable segmentation of the nuclear periphery has been achieved. Hence, a manual intervention from an expert was implemented in order to define two locations on the nuclear membrane: the closest point to the detected spot and on the opposite side of the nucleus in order to measure, respectively, the distance to the edge of the nucleus and its radius. The expert can also filter out any detection error and define the cell cycle.

### 2.9. Intensity Quantifications

Two different schemes are used to extract the spot intensities. The first, encapsulated inside the same Fiji macro for the spot localization, sums the intensities—minus the image background measured as its median intensity—of all the pixels comprised inside a three-by-three-by-three cube centered around the detection.

The other approach was preferred for data without the need to quantify the spot locations inside nuclei; thus, the signals inside vacuoles were not deemed problematic. It relies on qFOCI, a MATLAB application currently being developed to extract diverse information from yeast nuclei and the spots they may contain. With this method, the spots are fully segmented in 3D through multiple Hessian of Gaussian filters followed by an MSER detector [37] and their intensities are the total integrated pixel values. Finally, the previous Fiji macro was adapted to add cycle information for the cells of the detected spots and to discard any false detection.

### 2.10. Statistical Analysis

In order to assess the presence of statistically significant differences between two given yeast strains or incubation time, be it among the localizations or intensities, Kruskal–Wallis tests by ranks were performed [38]. This non-parametric method tests for the null hypothesis stating that the different sets of data originate for the same distribution. In the case of multiple comparisons, the *p*-values are corrected using Tukey’s range test [39]. All statistical analyses are performed with a significance level of 0.01 to reject the null hypothesis.

## 3. Results

### 3.1. An Inducible and Reversible System to Establish Gene Silencing at an Ectopic Locus

As mentioned above, *HML*-mediated gene repression is dependent on the proximity to telomeres [20]. When an *ADE2* gene flanked by the *HML* silencers is introduced at the *LYS2* locus located 339 kb away from telomeres (hereafter named *lys2::E-ADE2-I* locus), yeast colonies are white, indicative of *ADE2* expression [18]. However, introducing a LacI bound *lacO* array in proximity to this construct leads to the formation of pink colonies, indicative of *ADE2* repression [17]. Here we modified this strain, placing the GFP-LacIR, a galactose-resistant variant, under the control of the inducible *GALS* promoter to be able to tightly control its expression (Figure 1A). Cells bearing a wild-type or mutated allele of *ADE2* at its endogenous location are shown for comparison. As expected, in non-inducible conditions (glucose-containing medium) our inducible strain formed white colonies, similarly to a strain bearing no LacI gene (Figure 1B). In contrast, in inducible conditions (galactose-containing medium; Figure 1B) the inducible strain formed colonies as pink as the strain expressing GFP-LacIR under the control of the constitutive *HIS3* promoter. Consistent with our previous work [17], cells deleted for *SIR1, SIR3,* or *SIR4* formed white colonies confirming that the SIR complex repressed the *lys2::E-ADE2-I* locus, when the neighboring *lacO* array is bound by LacI proteins (Appendix A).

When pink colonies from the inducing plate were streaked on non-inducible medium, cells formed white colonies, indicating that the silencing at the ectopic locus is not maintained in the absence of LacI expression (Figure 1B, bottom row).

Thus, LacI binding to *lacO* in vicinity to the *E-ADE2-I* reporter gene induces a reversible SIR-dependent repressed state.

### 3.2. Perinuclear Anchoring Occurs independently of the Cell Cycle and Precedes Gene Silencing Establishment

As mentioned above, LacI induced silencing of the *lys2::lacO E-ADE2-I* reporter is accompanied by its association with the nuclear periphery [17]. We used our inducible system to monitor the subnuclear localization of the locus and the expression of the *ADE2* reporter gene upon LacI induction. Cells were first grown in raffinose-containing medium before adding galactose to the medium. Faint GFP foci, corresponding to LacI binding the *lacO* array, were visible in 51% of cells 30 min after galactose induction in all phases of the cell cycle (determined using the transmission images based on the absence or presence of a bud), suggesting that LacI binding does not require a passage through S phase. This percentage raised to 84% after 45 min, and above 95% after 90 min of induction (Figure 2A). Foci intensity increased rapidly before saturating after three hours of induction in both G1 and S-G2 phase cells (Figure 2B and Appendix A). Intensity distribution of GFP spots in the strain expressing constitutively the LacI protein under the *HIS3* promoter (here after the constitutive strain) is shown for comparison and is similar to the intensity observed after 45 min of induction (Kruskal–Wallis ANOVA test; *p* = 0.9). We noticed that in the constitutive strain, intensities of the GFP spots were lower in the G1 phase than in the S-G2 phase (Appendix A; *p* = 1 × 10^−10^), as expected, since the *lacO* array replicates during the S phase [40]. This difference was not observed after 30 min or 45 min of induction (*p* = 0.38 and 0.04 respectively), but it was statically significant after 90 min and even more obvious at later time points (*p* values ranging from 3 × 10^−6^ to 10^−10^).

We monitored the relative distance of the *lacO* array to the nuclear periphery using Nup49-mCherry to label the nuclear envelope. At early time points of induction, the spatial distribution of these foci in the nuclear space was slightly biased toward the nuclear periphery (Figure 2C and Appendix A) without detectable changes between 30 and 45 min (*p*-value > 0.99). In contrast, 90 min after LacI induction, we observed a strong shift toward the nuclear envelope (*p* = 4 × 10^−6^). This distribution was similar after 180 min of induction (>0.99) and even more peripheral after a long-term induction (1200 min, *p* = 3 × 10^−8^). At this stage the locus was more peripheral than in the strain expressing constitutively the LacI protein under the *HIS3* promoter (*p* = 3 × 10^−3^). We observed a similar behavior when scoring the position of the locus in cells in the S-G2 phase, indicating that this perinuclear anchoring is mostly independent of the cell cycle (Appendix A). In contrast, this perinuclear anchoring was not observed in cells without silencers at the *LYS*2 locus and expressing LacI under the weak constitutive *HIS3* promoter, when grown in the same medium (galactose containing medium). We thus conclude that perinuclear anchoring of the *LYS2* locus was not induced by growth in galactose medium but requires both the presence of silencer elements at this locus and LacI expression.

In parallel to the spatial localization of the locus, we monitored the establishment of the silencing at the *ADE2* reporter gene by measuring the *ADE2* mRNA levels at different time points after LacI induction (Figure 2D). To account for the change in expression induced by galactose addition independently of LacI induction, we calculated the ratio of mRNA levels between the LacI inducible strain and a strain that did not carry a LacI gene for each time point. Before induction, *ADE2* mRNA levels were 6.5-fold higher than in the strain with constant LacI expression. We observed a first 25% decrease in mRNA levels after 90 min of LacI induction, after which we observed little variations up to 360 min after induction. It was only after a long-term induction (1200 min) that mRNA levels reached the levels of the constitutive strain. Such a delay between LacI binding that reached the levels sufficient to induce silencing in the constitutive strain after 45 min of induction and the decrease in mRNA levels cannot be explained by the half-life of the *ADE2* mRNA (which estimation varies between 13 and 40 min [41,42,43,44]).

Thus, complete repression required a long-term induction, although maximum LacI induction and binding were reached after 180 min. In contrast, association with the nuclear periphery was to the level of the constitutively silent loci after 90 min of induction and further increased upon long-term induction.

### 3.3. Sir3 and Sir4 Accumulate over Several Cell Cycles following LacI Induction

We next assessed the recruitment of Sir3 along the *lys2::E-ADE2-I* locus by chromatin immunoprecipitation (ChIP) analyzed by quantitative PCR, using primers located along the locus at different time points after LacI induction (Figure 3A). We previously showed a basal recruitment of Sir3 and Sir4 at the *E* silencer and at the *ADE2* reporter gene flanked by the *HML* silencers, with or without an unbound *lacO* array located 1.9 kb upstream of the silencers (Dubarry et al., 2011). Consistently we observed a low level of Sir3 and Sir4 at these sites before LacI induction (Figure 3A and Appendix A). Upon induction, Sir3 and Sir4 binding gradually increased all along the locus.

The fastest and highest recruitment was observed at the *E* silencer (Figure 3A,B and Appendix A), suggesting that Sir3 and Sir4 are first recruited at this site before spreading on the flanking regions. However, after 270 min of induction Sir3 enrichment was still below the one of the strains with constitutive LacI expression, this level being reached only after a long-term induction (Figure 3A,B). Probing Sir3 binding after 20 h, 48 h, and 72 h of induction revealed that Sir3 recruitment did not show significant variation after 20 h (Appendix A). 

Focusing on the individual amplicons, we observed that Sir3 recruitment increased similarly at the *lacO* array and at the *E* silencer during the first 3 h of induction and increased abruptly at the *E* silencer after this time point, while Sir3 increased more progressively at the *ADE2* gene over the time course (Figure 3B). These data suggest that the SIR complex is recruited both at the *lacO* array and the silencer at early time points before spreading on the *ADE2* gene.

The kinetics of Sir3 and Sir4 recruitment are in good agreement with the slow repression observed in Figure 2 and the kinetics of H4K16 deacetylation (Figure 3C and Appendix A). Indeed, H4K16 ChIP signal showed little variations during the first 180 min after LacI induction in the body of the reporter gene and reached the levels of the constitutive strain only after a long-term induction (Figure 3C and Appendix A). In contrast, we noticed an abrupt decrease in H4K16 acetylation along the *lacO* array, during the first 90 min following LacI induction (Appendix A), even when normalizing the H4K16Ac ChIP signal by nucleosome occupancy (Appendix A).

Altogether, our data show that heterochromatin formation and silencing establishment are slow processes taking place over several cell cycles. This is in contrast with the kinetic of perinuclear anchoring that occurs mainly during the first 90 min following LacI induction.

### 3.4. Perinuclear Anchoring Is Partly Sir3 Independent

The difference in kinetic of perinuclear anchoring and silencing factor recruitment suggests that perinuclear anchoring of the *lys2::E-ADE2-I* locus could occur independently of heterochromatin formation upon LacI binding. To test this hypothesis, we monitored the distance of this locus to the nuclear periphery in *SIR3* or *SIR4* deleted strains (Figure 4A and Appendix A). In the *sir3∆* strain, the localization was similar to the wt strain, when monitored 30–45 min after LacI induction (*p* = 0.96). After 3 h of induction the locus was shifted toward the nuclear periphery to a similar extent than in the wt strain (*p* = 0.9), but no further changes in localization were observed after a long-term induction in both *sir3∆* and *sir4∆* strains. Similar results were obtained in G1 or S-G2 cells (Appendix A).

We thus observed an SIR-independent anchoring of the locus to the nuclear periphery following LacI induction. We next tested whether this perinuclear anchoring was also independent of the presence of a neighboring silencer element.

In the absence of silencers, the *LYS2* locus also showed a more peripheral localization upon high LacI expression compared with a strain expressing the LacI protein under the weaker constitutive *HIS3* promoter grown in the same medium (Figure 4B, *p* = 3.8 × 10^−9^). The only difference between these two strains is the amount of LacI expression and binding to the *lacO* array, the median intensity of the *lacO* spot being 3.7-fold higher in the induced strain than in the constitutive strain (Figure 4C,D). Therefore, a high level of LacI binding is sufficient to induce a shift of the locus to the nuclear periphery independently of Sir3, Sir4, or the presence of silencer elements. However, this LacI induced Sir3-independent perinuclear anchoring is weaker than the one observed when the locus is silenced (*p* = 5 × 10^−4^). We thus conclude that the perinuclear anchoring of the *lys2::lacO E-ADE2-I* locus, following LacI binding to the *lacO* array, is partially independent of heterochromatin establishment.

### 3.5. LacI Binding Induces a Mec1 Dependent H2A-P

We hypothesized that the Sir3-independent change in localization induced upon LacI binding could result from chromatin stress induced by the LacI binding to the *lacO* array. We thus set out to monitor the presence of histone H2A phosphorylation, a mark associated both with silent chromatin and DNA damage [45,46], at the *lys2::lacO E-ADE2-I* locus following LacI induction. To account for potential variation in H2A occupancy, we normalized the H2A phosphorylation (H2A-P) ChIP signal against total H2A ChIP performed on the same samples.

The H2A-P/H2A signal increased with time after induction over the locus, with a faster and higher increase at the *E* silencer, suggesting that H2A was first phosphorylated there before the mark spread on the flanking regions (Figure 5A and Appendix A).

We next tested the respective contribution of the two H2A kinases Mec1 and Tel1 in this phosphorylation event since Tel1 is the main kinase responsible for H2A-P at telomeres ([45] and Appendix A), whereas Mec1 is the main kinase acting at processed DSB [47]. While deleting *TEL1* had no significant effect on the kinetic and level of H2A-P, deleting *MEC1* led to a reduction in H2A-P (Figure 5B) upon LacI induction compared to the wild-type strain. However, we still observed a slow accumulation of H2A-P at the *E* silencer in the absence of Mec1. This suggests that although Mec1 is the main kinase responsible for H2A-P at the *E* silencer upon LacI induction, Tel1 can at least partially substitute Mec1 activity.

### 3.6. H2A-P Induced by LacI Binding Is Sir3 Independent but Is Stabilized by Sir3 Spreading

As mentioned above, genome-wide studies showed a strong correlation between H2A-P and silent chromatin [5,45,46].

The significant increase in H2A-P at the *E* silencer as soon as 45 min after induction, before any significant change in localization or in SIR recruitment occurred, indicated that this phosphorylation event could be SIR independent. We tested this hypothesis by monitoring H2A-P/H2A in a *SIR3* deleted strain upon LacI induction. We observed a transient phosphorylation of H2A at the *E* silencer and the *lacO* array in this strain (Figure 5C and Appendix A). This Sir3 independent, LacI induced H2A-P peaked 90 min after induction before slowly decreasing to reach basal levels upon a long-term induction at the *E* silencer and a two-fold enrichment at the *lacO* array (Figure 5C and Appendix A). Thus, while the initiation of H2A-P is Sir3 independent, its accumulation depends on the presence of Sir3. These data suggest that H2A phosphorylation is an early event induced by LacI binding at the *lacO* array, and that this mark is later stabilized by Sir3 binding to nucleosomes. In good agreement with this hypothesis, we also observed a phosphorylation of H2A after 90 min of LacI induction at the same distance from the *lacO* array in the strain without silencer (Appendix A).

H2A phosphorylation being the first event that we could detect at the silencer, we wondered whether this modification could be responsible for the recruitment of the SIR complex. We thus assayed the H2A-S129A (*hta-S129A*) mutant that could not be phosphorylated by the PI3 kinases Mec1 and Tel1 (Downs et al. 2000), for Sir3 recruitment. Although we observed a slight defect of Sir3 recruitment at the *E* silencer after a long-term LacI induction, this was not the case at the *ADE2* gene (Figure 5D). Consistently, we did not observe any defect in silencing establishment upon LacI induction, neither in this strain nor in the *MEC1* or *TEL1* deleted strain (Figure 5E). Similarly, Sir3 recruitment was reduced in the H2A-S129A mutant strain 0.2 kb from the telomere 6R but unaffected in further distant subtelomeric regions (Appendix A). Therefore, H2A-P may have a small contribution at the nucleation step but did not impact Sir3 spreading and the resulting gene silencing.

We next tested the impact of this modification on the perinuclear anchoring of the locus. Distributions of the locus in *mec1∆* or H2A-S129A expressing strains were similar to the distribution in the wild-type strain (Figure 5F, *p* = 0.76). In the absence of Sir3, the localization of the locus was not significantly different in the absence or presence of this mutation, although we noticed a higher variability of the data in the double mutant (*p* = 0.8; Appendix A). Thus, H2A phosphorylation is not required for the perinuclear anchoring of the *lys2::lacO E-ADE2-I* locus upon LacI binding.

In summary, the H2A phosphorylation event observed 45 min after LacI binding on the *lacO* array is independent on Sir3 recruitment, and reciprocally, this modification is not required for Sir3 recruitment or the perinuclear anchoring of the locus. However, Sir3 recruitment that occurs later after LacI induction (starting at 90 min) allows the stabilization of this mark, which then accumulates over the long-term induction.

### 3.7. Sir3 Spreading Counteracts LacI-Induced Nucleosome Eviction and Limits LacI Binding

Plotting H2A occupancy along the locus at different times after induction revealed a progressive decrease in H2A ChIP signal over the *lacO* array, from 45 min to 3 h after induction. At this time, H2A enrichment (over our reference locus, *OGG1)* was 3-fold lower than before LacI induction. However, we observed a slight but reproducible increase in H2A occupancy after long-term induction (Figure 6A,B). We wondered whether this late re-association of H2A could be related to the SIR spreading observed after long-term induction (Figure 3). Indeed, in the absence of Sir3, H2A eviction was stronger than in the wild-type strain 3 h after induction and remained stable after long-term LacI induction (Figure 6B). In the constitutive strain expressing a lower level of LacI, we observed a 20 to 25% decrease in H2A occupancy as compared with the signal obtained in the absence of LacI binding (Figure 6C). Again, in the absence of Sir3, the eviction was stronger, with H2A occupancy at the *lacO* array corresponding to 30% of the occupancy of the unbound array. A similar level of H2A ChIP signal was observed in a wt strain expressing the LacI under the *HIS3* promoter but without silencer flanking the *lacO* array (Appendix A). These results suggest that Sir3 spreading from the silencer stabilizes nucleosomes that are removed by LacI binding in the absence of Sir3.

It is noteworthy that canonical sites of SIR recruitment are also associated with nucleosome-poor regions such as the TG repeats, subtelomeric sequences, and the *HM* silencers. Indeed, H2A signal is rather low immediately adjacent to telomere 6R compared to other genomic sites (60% of the reference signal), and it is even lower in the absence of Sir3 (Figure 6D), indicating that Sir3 spreading stabilizes the nucleosome, in good agreement with previous work [48,49,50].

We then asked whether Sir3 spreading could prevent LacI binding at the *lacO* array. We thus compared the intensity of the GFP-LacI spot neighboring the *E-ADE2-I* reporter in wild-type or *sir3∆* strain expressing the LacI protein under the constitutive *HIS3* promoter (Figure 6E). We observed that the intensity of the spots in G1 cells was higher in the absence of Sir3 (*p* = 10^−10^, Figure 6F). Similarly, GFP-LacI spots were brighter in the absence than in the presence of a neighboring silencer element (Appendix A). We thus conclude that Sir3 spreading impairs LacI binding.

All together these data indicate that Sir3 stabilizes nucleosome and prevents the access of DNA binding factor to DNA.

## 4. Discussion

Here we set up an inducible system to monitor silencing establishment at an ectopic locus without affecting SIR protein expression or activity.

In this experimental system, silencing establishment is induced by the binding of the LacI protein on a *lacO* array located 1.9 kb away from a reporter of ectopic silencing consisting of the *ADE2* gene flanked by the *HML* silencers inserted at the *LYS2* locus [17,18].

Using this system, we observe that LacI binding is accompanied by nucleosome eviction at the *lacO* array, together with the phosphorylation of H2A within the *lacO* array and the neighboring nucleosomes. This is followed by the perinuclear anchoring of the locus and the progressive recruitment of the SIR complex at the *E* silencer from where it spreads on the flanking regions, including the *lacO* array and the *ADE2* reporter gene leading to its repression after long-term induction. Another consequence of SIR spreading is the accumulation of H2A-P on the underlying nucleosomes. While the SIR recruitment and the initial phosphorylation of H2A are independent events, Sir3 recruitment allows the stabilization of H2A-P. Finally, Sir protein recruitment stabilizes nucleosomes and counteracts LacI binding.

### 4.1. LacI Binding Induces H2A Phosphorylation and SIR Recruitment Independently

The phosphorylation of H2A is the earliest event that we could detect upon LacI binding at the *lacO* array. This histone modification is triggered by the PI3 kinases Mec1 and Tel1 that are activated upon DNA damage, suggesting that LacI binding induces a DNA or chromatin stress that is sensed by the checkpoint machinery. This chromatin stress could result from the abrupt eviction of nucleosomes upon LacI binding. We observed that this LacI binding induced phosphorylation of H2A depends on both Mec1 and Tel1 with a preponderant impact of Mec1. However, we could not detect the phosphorylation of the downstream effector Rad53 (Appendix A) indicating a mild activation of the PI3 kinases.

Both Tel1 and the DNA repair factor Mre11 were shown to interact with members of the SIR complex in the two-hybrid system and to promote silencing when artificially tethered to a defective silencer [51]. Hence, Tel1 and Mre11 may trigger silencing through the direct recruitment of the SIR complex. Furthermore, both Mre11- and Tel1-mediated silencing was at least partially dependent on H2A phosphorylation. However, none of these factors is required for silencing establishment in our system (Figure 5). This is consistent with the fact that these factors are also not required for silencing at *HM* loci, despite being sufficient in Reference [51].

Genome wide, H2A-P is found enriched at SIR bound loci including subtelomeres and *HM* loci, but also smaller regions located on chromosome arms [5,45,46], and this enrichment decreased in the absence of Sir3. Similarly, we observed a strong decrease in H2A-P in the absence of Sir3 after long-term LacI induction. However, the phosphorylation of H2A observed immediately after LacI binding is Sir3 independent. H2A-P and Sir3 recruitment are thus independent events induced by LacI binding at the *lacO* array. Still, Sir3 spreading stabilizes this otherwise transient mark, leaving a scar of the initial chromatin stress. A similar mechanism could explain the presence of H2A-P at silent loci since natural silencers are made of binding sites for proteins able to destabilize nucleosomes such as Rap1, Abf1, or Orc1. Their binding could cause a chromatin stress, leading to H2A-P that is then stabilized by Sir3 binding. Whether H2A-P and/or Sir3 helps stabilizing these loci will be interesting to study in the future.

### 4.2. Silencing Establishment Occurs over Several Cell Cycle

In contrast to H2A-P, SIR recruitment and silencing establishment is a slow process, with Sir3 and Sir4 binding increasing significantly after 90 min and accumulating over several cell cycles, leading to gene silencing only after long-term induction (20 h). This slow establishment of silencing is in contrast with previous reports monitoring silencing establishment at *HM* loci [24,25,26,27,28,30,31,32,50]. This difference could be explained by several factors: (i) the chromosomal position of the reporter studied (euchromatic regions 339 kb away from *Tel2R* in our system versus subtelomeric *HML* loci); (ii) the strength of the gene promoter [30]; (iii) the method used to induce silencing establishment. Indeed, in our system silencing is induced while silencing is already established at other loci, whereas previous studies used conditional or inducible alleles of *SIR3,* thus affecting all silent loci simultaneously. These factors are not mutually exclusive and could even reinforce each other. Because Sir3 is available in a limited amount for silencing establishment, it might be more difficult to recruit Sir3 molecules when they are sequestered at other loci, and the distance to these loci might decrease further the probability to encounter Sir3 molecules [19,20,21,22].

### 4.3. Anchoring at the Nuclear Periphery Precedes Silencing Establishment

Silent loci are enriched at the nuclear periphery, whereas the *LYS2* locus, when not silent, shows an almost random nuclear localization [17]. Here we observed a progressive accumulation of the locus at the nuclear periphery that precedes silencing establishment. Consistently, this perinuclear anchoring is partially independent of the SIR complex. How LacI binding triggers this change in nuclear localization is not clear. This change in localization that is observed both in G1 and S-G2 phases of the cell cycle, independently of H2A-P, Tel1, or Mec1, requires a high level of LacI expression. This high level of LacI binding results in nucleosome eviction, suggesting that a large nucleosome-free region might be responsible for this relocalization at the nuclear periphery. This SIR-independent perinuclear anchoring could contribute to silencing establishment by bringing the locus in proximity to SIR dense regions, as proposed earlier [21,22].

### 4.4. Possible Mechanisms for Silencing Establishment

What is the mechanism leading to SIR recruitment upon LacI binding? We previously proposed that the replication stress arising from tight DNA–protein interactions could favor heterochromatin formation [17]. Here we show that LacI binding is accompanied by a decrease in histone occupancy along the *lacO* array. Binding and unbinding of the LacI protein could increase the turnover of nucleosomes independently of DNA replication, thus leading to the incorporation of unmodified histones that are more permissive to Sir3 binding. In particular, H3K79 tri-methylation, which is widespread in euchromatin, counteracts Sir3 spreading and prevents silencing establishment [5,32,52]. Given the lack of enzyme to erase this histone mark, its removal can be achieved only through turnover of the histones that bear it. Histone turnover occurs mainly during the S phase and was proposed to be a critical cell-cycle-regulated step in silencing establishment [32]. Our data suggest that the dynamics of LacI binding and unbinding could also stimulate histone turnover on the *lacO* array.

Nucleosome-depleted regions (NDR) were recently shown to improve silencing establishment and by favoring nucleosome positioning when the distance between two NDRs allows a regular positioning of nucleosomes [53]. However, this effect decreases with the distance between the two NDRs and is, thus, unlikely to be at play in our system where the distance between the *E* silencer and the *lacO* array is 1960 bp. Yet, it would be interesting to test whether shortening this distance to a length that could not be evenly divided into nucleosomes would prevent silencing establishment.

As mentioned above, the perinuclear anchoring of the locus that occurs independently of SIR recruitment could also contribute to SIR recruitment by bringing the locus in proximity to the pool of SIR proteins associated with silent loci enriched at the nuclear periphery. In turn, SIR recruitment reinforces the perinuclear anchoring of the locus, feeding a positive loop that increases both SIR recruitment and anchoring at the nuclear periphery.

### 4.5. SIR Spreading Stabilizes Nucleosome and Limits the Access to DNA

The use of our inducible system to study the establishment of silencing at an ectopic locus revealed a positive role for the SIR complex in the stabilization of nucleosomes in vivo. This was also observed at the natural subtelomere 6R where the sequence immediately adjacent to the telomeric repeats shows a low occupancy in wild-type cells, which is further reduced in the absence of Sir3. A similar effect of heterochromatin in stabilizing nucleosomes has also been reported in *S. pombe* [54,55]. This stabilization could be a direct consequence of the SIR complex spreading on nucleosomes or an indirect effect owing from the protection of nucleosomes against histone modifiers and remodelers or to a steric hindrance preventing the binding of DNA binding protein. Our observation that SIR spreading at the *lacO* array reduces LacI binding indicates that SIR bound to nucleosomes occlude the access of DNA binding protein to DNA. This is in good agreement with in vitro studies suggesting that association of the SIR complex to chromatin results in the formation of a fiber characterized by Sir3-dependent stabilization of nucleosomes and occlusion of DNA linkers [56,57].

In conclusion, using our inducible system, we showed that a massive binding of LacI protein leads to nucleosome eviction, inducing a chromatin stress as indicated by the phosphorylation of H2A at this locus and its anchoring at the nuclear periphery. This is followed by the recruitment of the SIR complex, which in turn stabilizes nucleosomes, H2A-P, and the perinuclear anchoring of the locus and prevents the access of DNA binding protein to DNA.

Although artificial, this system may reveal a mechanism at play at natural genomic loci. Indeed, natural sites of SIR recruitment such as *HM* silencers or telomeric repeats are made of combinations of binding sites for protein that are strong nucleosome-displacing factors (NDFs) [6]. Moreover, binding sites for other NDFs such as Ume6, Reb1, or TFIIIC can cooperate with silencer elements to strengthen silencing [6,17,53]. Further research will be needed to decipher the precise mechanism by which nucleosome-depleted regions can lead to silencing establishment.

## Figures and Tables

**Figure 1 cells-10-02810-f001:**
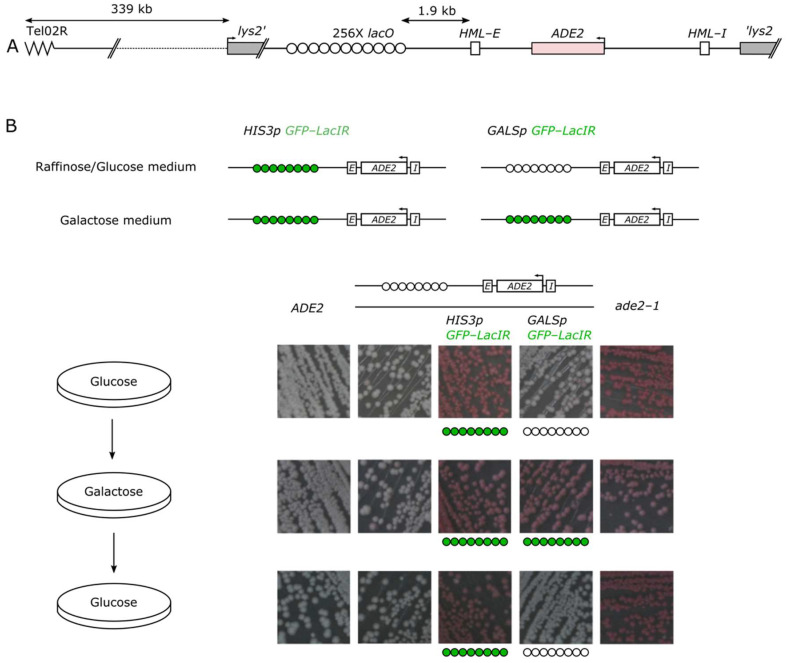
An inducible and reversible system to establish silencing at an ectopic locus. (**A**) Schematic representation of the construct used to study silencing establishment. A *lacO* array and the *ADE2* reporter gene, flanked by *HML* silencers *E* and *I*, were inserted at the *LYS2* locus and referred to as the *lys2::lacO E-ADE2-I* locus. (**B**) Assessing the inducibility and reversibility of the system. The GFP-LacIR protein (R for resistant to galactose) was expressed either under the constitutive *HIS3* promoter (constitutive strains) or under the inducible *GALS* promoter (inducible strains). We followed the silencing of the *ADE2* gene by recording the color of the colonies streaked on plates (colonies are white when *ADE2* is expressed and pink when it is repressed). From left to right, pictures of a wild-type (WT) *ADE2* strain as a control (yAT126), a strain bearing the *lacO E-ADE2-I* construct at *LYS2* (yAT2001), a strain bearing the *lacO E-ADE2-I* construct at *LYS2* and expressing the integrated GFP-LacIR under the constitutive *HIS3* promoter (yAT2059) or under the inducible *GALS* promoter (yAT2078), and the WT *ade2-1* strain unable to produce adenine as a control (yAT1). Strains were streaked on glucose (repressive condition for the *GALS* promoter), then from glucose to galactose (inducing conditions for the *GALS* promoter) and back to glucose plates.

**Figure 2 cells-10-02810-f002:**
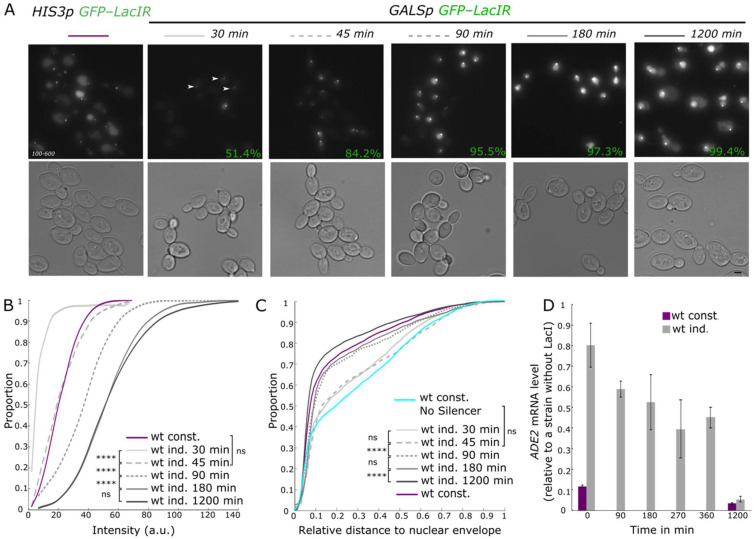
Perinuclear anchoring occurs independently of the Cell Cycle and precedes gene silencing establishment. (**A**) Representative fluorescent images (maximal projection of a Z-stack) and their corresponding transmission images of strains bearing the *lys2::lacO E-ADE2-I* locus and expressing GFP-LacIR either under the constitutive *HIS3* promoter (wt const. yAT2059) or under the inducible *GALS* promoter (wt ind. yAT2078) at 30 min, 45 min, 90 min, 180 min, and 1200 min after galactose induction. Percentage of cells with one spot is given for each time point after galactose induction of the GFP-LacIR protein. Bar, 2 µm. (**B**) Cumulative distributions of the intensity of GFP foci in G1 cells, in strains bearing the *lys2::**lacO E-ADE2-I* locus and expressing GFP-LacIR either under the constitutive *HIS3* promoter (wt const. yAT2059) or under the inducible *GALS* promoter (wt ind. yAT2078) at 30 min, 45 min, 90 min, 180 min, and 1200 min after galactose induction of the GFP-LacIR protein. For statistics: *p* > 0.01 is non-significant (ns), *p* ≤ 0.00001(****) and see Appendix A for statistics. (**C**) Cumulative distributions of the relative distance of *lys2::**lacO E-ADE2-I* or *lys2::**lacO* locus to the nuclear periphery in G1 cells, in strains expressing GFP-LacIR either under the constitutive *HIS3* promoter (wt const. yAT2059 and wt cont. No Silencer yAT3420) grown in galactose-containing medium or under the inducible *GALS* promoter (wt ind. yAT2078) at 30 min, 45 min, 90 min, 180 min, and 1200 min after galactose induction of the GFP-LacIR protein. *p* > 0.01 is non-significant (ns), *p* ≤ 0.00001(****) and see Appendix A for statistics. (**D**) Graph showing the levels of *ADE2* mRNA (RT-qPCR) isolated from strains bearing *lys2::**lacO E-ADE2-I* locus and expressing GFP-LacIR either under the constitutive *HIS3* promoter (wt const. yAT2059) or under the inducible *GALS* promoter (wt ind. yAT2078) in raffinose (t = 0, no induction) and after 90 min, 180 min, 270 min, 360 min, and 1200 min after galactose induction. To account for changes in gene expression related to growth conditions independently of LacI induction, *ADE2* mRNA levels were normalized to the *ADE2* mRNA levels of a strain not expressing the LacI protein (data are shown as mean ± s.e.m.; see Appendix A for the number of experiments).

**Figure 3 cells-10-02810-f003:**
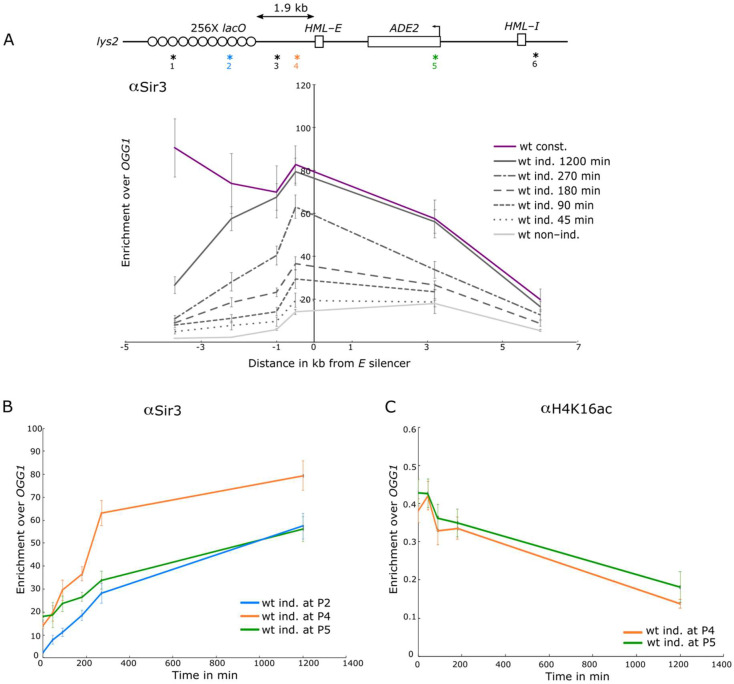
Heterochromatin formation takes place over several cell cycles. (**A**) Sir3 occupancy along the of *lys2::**lacO E-ADE2-I* locus, probed by ChIP-qPCR using an anti-Sir3 antibody developed by us (Ruault et al., 2011), in strains expressing GFP-LacIR either under the constitutive *HIS3* promoter (wt const. yAT2059) or under the inducible *GALS* promoter (wt ind. yAT2078) in raffinose (no induction), or after 45 min, 90 min, 180 min, 270 min, and 1200 min of galactose induction of the GFP-LacIR protein. The data were normalized over *OGG1* (shown as mean ± s.e.m.). (P1), (P2), (P3) (P4) amplicons are respectively located at 3.7 kb, 2.2 kb, 1 kb, and 0.5 kb from the left side of the *E* silencer, and (P5) and (P6) amplicons are located at 3.2 kb and 6 kb from the right side of the *E* silencer. Amplicon positions are localized on the scheme of the locus with an asterisk and their respective number (see Appendix A for the number of experiments). (**B**) Sir3 enrichment over time at three particular sites of the *lacO E-ADE2-I* construct inserted at the *LYS2* locus: at the *lacO* (P2), nearby the *E* silencer (P4), and in the body of *ADE2* (P5) obtained by plotting data from Figure 3A. (**C**) H4K16 acetylation occupancy nearby the *E* silencer (P4) and in the body of *ADE2* (P5) in a strain bearing the *lys2::lacO E-ADE2-I* locus, probed by ChIP-qPCR using an anti-acetyl-histone H4 (Lys16) antibody, in strains expressing GFP-LacIR under the inducible *GALS* promoter (wt ind. yAT2078) in raffinose (no induction), or after 45 min, 90 min, 180 min, and 1200 min of galactose induction of the GFP-LacIR protein. The data were normalized over *OGG1* (shown as mean ± s.e.m.; see Appendix A for the number of experiments).

**Figure 4 cells-10-02810-f004:**
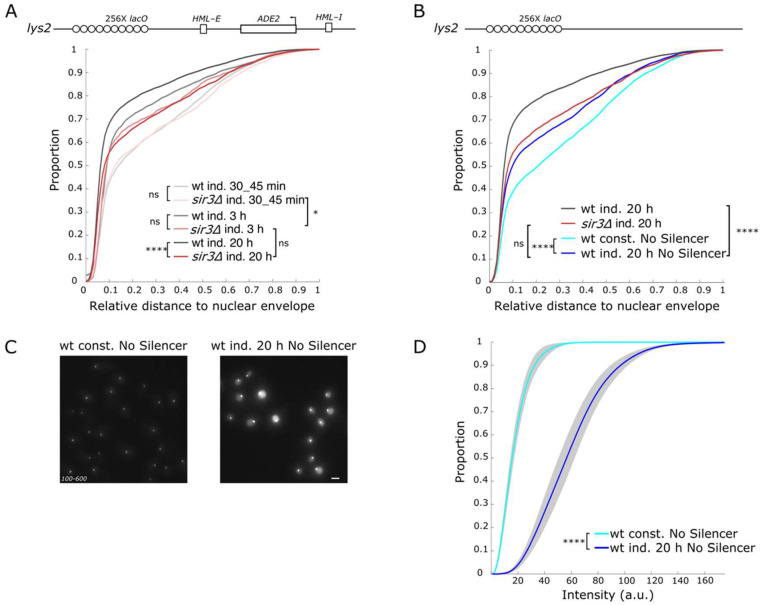
Perinuclear anchoring is partly Sir3 independent. (**A**) Cumulative distributions of the relative distance of the *lys2::lacO E-ADE2-I* locus to the nuclear periphery, in G1 cells, in strains expressing GFP-LacIR under the control of the *GALS* promoter in the WT (wt ind., yAT2078), and in a *sir3∆* strain (*sir3∆ ind.,* yAT2370) after 30–45 min, 3 h, and 20 h of galactose induction. *p* > 0.01 is non-significant (ns), *p* ≤ 0.01 (*), *p* ≤ 0.00001(****) and see Appendix A for statistics. (**B**) Cumulative distributions of the relative distance of *lys2::**lacO E-ADE2-I* locus to the nuclear periphery in G1 cells, in strains expressing GFP-LacIR under the inducible *GALS* promoter in the WT (wt ind. yAT2078), and in a *sir3∆* (*sir3∆ ind.,* yAT2370) strains after 20 h of galactose induction. Cumulative distributions of the relative distance of the *lys2::lacO* array locus (No Silencer) to the nuclear periphery in G1 cell, in strains expressing GFP-LacIR either under the inducible *GALS* promoter (wt ind. No Silencer, yAT3471) after 20 h of galactose induction or under the constitutive *HIS3* promoter grown in galactose (wt const. No Silencer, yAT3420). *p* > 0.01 is non-significant (ns), *p* ≤ 0.00001 (****) and see Appendix A for statistics. (**C**) Representative fluorescent images (maximal projection of a Z-stack) of strains bearing the *lys2::**lacO* array locus (No Silencer) and expressing GFP-LacIR either under the constitutive *HIS3* promoter (wt const. No Silencer yAT3420) or under the inducible *GALS* promoter (wt ind. No Silencer yAT3471) at 20 h after galactose induction. Bar, 2 µm. (**D**) Cumulative distributions of the intensity of GFP foci in G1 cells, in strains bearing a *lys2::**lacO* array locus (No Silencer) and expressing GFP-LacIR either under the constitutive *HIS3* promoter (wt const. No Silencer yAT3420) or under the inducible *GALS* promoter (wt ind. No Silencer yAT3471) at 20 h after galactose induction. *p* > 0.01 is non-significant (ns), *p* ≤ 0.01 *p* ≤ 0.00001(****) and see Appendix A for statistics.

**Figure 5 cells-10-02810-f005:**
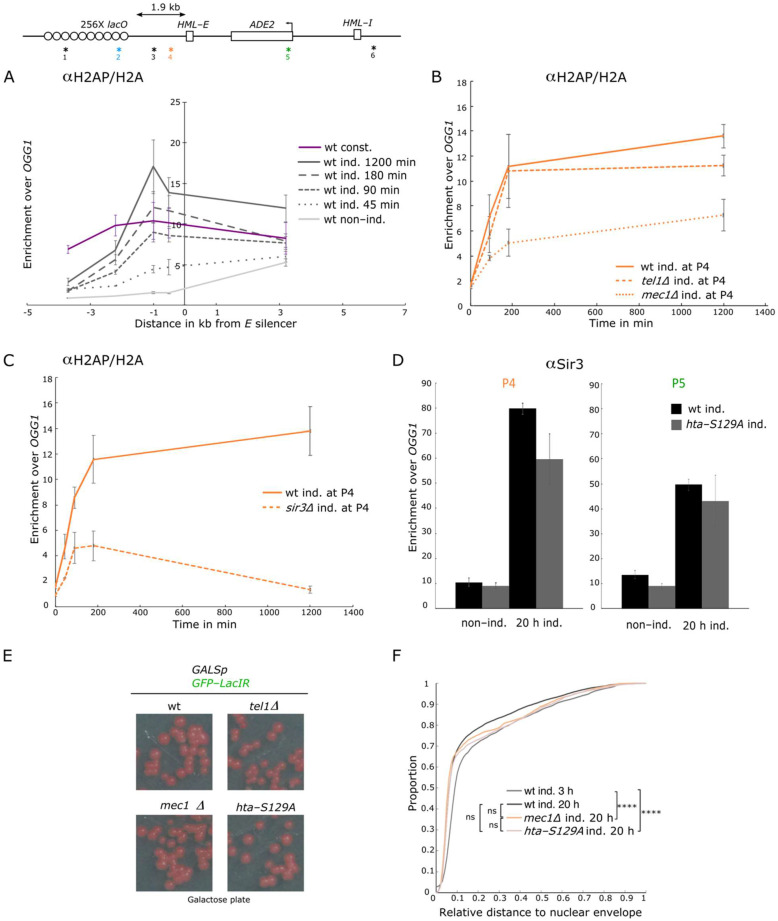
Mec1-dependent H2A phosphorylation induced by LacI binding and stabilized by Sir3 spreading. (**A**) H2A-S129 phosphorylation occupancy along the *lys2::**lacO E-ADE2-I* locus probed by ChIP-qPCR using an anti-H2A Phospho-S129 antibody in strains expressing GFP-LacIR either under the constitutive *HIS3* promoter (wt const. yAT2059) or under the inducible *GALS* promoter (wt ind. yAT2078) in raffinose (no induction), or after 45 min, 90 min, 180 min, and 1200 min of galactose induction. The data were normalized over *OGG1* and histone H2A ChIP signal (shown as mean ± s.e.m.; see Appendix A for the number of experiments). (P1), (P2), (P3) (P4) amplicons are respectively located at 3.7 kb, 2.2 kb, 1 kb, and 0.5 kb from the left side of the *E* silencer, and (P5) and (P6) amplicons are located at 3.2 kb and 6 kb from the right side of the *E* silencer, amplicon positions are localized with an asterisk and their respective number on the scheme of the locus. (**B**) H2A-S129 phosphorylation enrichment over time nearby the *E* silencer (P4) in strains bearing the *lys2::**lacO E-ADE2-I* locus, probed by ChIP-qPCR using an anti-H2A Phospho-S129 antibody, in strains expressing GFP-LacIR under *GALS* promoter in the WT (wt ind., yAT2078), in a *tel1∆* (*tel1∆ ind,.* yAT2314), and in a *mec1∆* (*mec1∆ ind.,* yAT2904) strains in raffinose (no induction), or after 90 min, 180 min, and 1200 min of galactose induction. The data were normalized over *OGG1* and histone H2A ChIP signal (shown as mean ± s.e.m.; see Appendix A for the number of experiments). (**C**) H2A-S129 phosphorylation enrichment over time nearby the *E* silencer (P4) in strains bearing the *lys2::**lacO E-ADE2-I* locus, probed by ChIP-qPCR using an anti-H2A Phospho-S129 antibody, in strains expressing GFP-LacIR under *GALS* promoter in the WT (wt ind., yAT2078), and in a *sir3∆* (*sir3∆ ind,.* yAT2370) strain in raffinose (no induction), or after 45 min, 90 min, 180 min, and 1200 min of galactose induction. The data were normalized over *OGG*1 and histone H2A ChIP signal (shown as mean ± s.e.m.; see Appendix A for the number of experiments). (**D**) Sir3 enrichment probed by ChIP-qPCR using Sir3 antibody nearby the *E* silencer (P4) and at *ADE2* (P5) in strains bearing the *lys2::**lacO E-ADE2-I* locus, and expressing GFP-LacIR under the *GALS* promoter in the WT (wt ind., yAT2078), and in a *hta-S129A* (*hta-S129A* ind, yAT2965) strains in raffinose (no induction), and 20 h of galactose induction. The data were normalized over *OGG1* (shown as mean ± s.e.m.; see Appendix A for the number of experiments). (**E**) Pictures of yeast colonies bearing the *lys2::**lacO E-ADE2-I* locus and expressing the GFP-LacIR under the *GALS* promoter in the WT (yAT2078), in a *tel1∆* (yAT2314), in a *mec1∆* (yAT2904), and in a *hta-S129A∆* (yAT2965) strains streaked on galactose plate. (**F**) Cumulative distributions of the relative distance of *lys2::**lacO E-ADE2-I* locus to the nuclear periphery in G1 cells, in strains expressing GFP-LacIR under the *GALS* promoter in the WT (wt ind., yAT2078) strain after 3 h and 20 h of galactose induction, in the *mec1∆* (*mec1∆ ind.,* yAT2904) and in the *hta-S129A∆* (*hta-S129A* ind, yAT2965) strains after 20 h of galactose induction. *p* > 0.01 is non-significant (ns), *p* ≤ 0.00001(****) and see Appendix A for statistics.

**Figure 6 cells-10-02810-f006:**
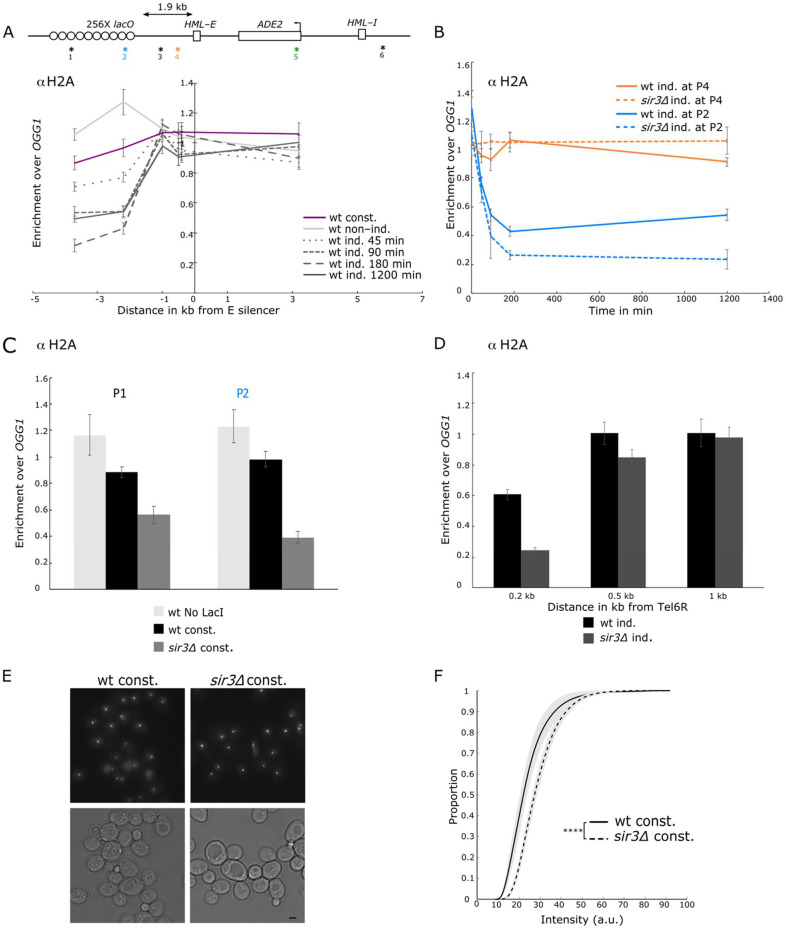
Sir3 spreading counteracted LacI-induced nucleosome eviction. (**A**) H2A occupancy along the of *lys2::**lacO E-ADE2-I* locus probed by ChIP-qPCR using an anti-H2A antibody in strains expressing either the GFP-LacIR under the constitutive *HIS3* promoter (wt const., yAT2059) or under the *GALS* promoter (wt ind., yAT2078) in raffinose (no induction), or after 45 min, 90 min, 180 min, and 1200 min of galactose induction. (P1), (P2), (P3) (P4) amplicons are respectively located at 3.7 kb, 2.2 kb, 1 kb, and 0.5 kb from the left side of the *E* silencer, and (P5) and (P6) amplicons are located at 3.2 kb and 6 kb from the right side of the *E* silencer, amplicon positions are localized with an asterisk and their respective number on the scheme of the locus The data were normalized over *OGG1* (shown as mean ± s.e.m.; see Appendix A for the number of experiments). (**B**) H2A enrichment over time at the *lacO* (P2), and nearby the *E* silencer (P4), in a strain bearing the *lys2::lacO E-ADE2-I* locus, probed by ChIP-qPCR using an anti-H2A antibody in strains expressing the GFP-LacIR under the inducible *GALS* promoter in the WT (wt ind., yAT2078) and in a *sir3∆* (*sir3∆ ind.,* yAT2370) strain, in raffinose (no induction), or after 45 min, 90 min, 180 min, and 1200 min of galactose induction. The data were normalized over *OGG1* (shown as mean ± s.e.m.; see Appendix A for the number of experiments). (**C**) H2A occupancy at the *lacO* (P1 and P2) in a strain bearing the *lys2::lacO E-ADE2-I,* probed by ChIP-qPCR using an anti-H2A antibody, in a strain without LacI expression (wt No LacI, yAT2001) or in strains expressing GFP-LacI under the control of the constitutive *HIS3* promoter in the WT (wt const., yAT2000) and in a *sir3∆* (*sir3∆ const.*, yAT2156). The data were normalized over *OGG1* (shown as mean ± s.e.m.; see Appendix A for the number of experiments). (**D**) H2A occupancy at 0.2 kb, 0.5 kb, and 1 kb from the end of telomere 6R in strains bearing the *lys2::lacO E-ADE2-I* locus, probed by ChIP-qPCR using an anti-H2A antibody, and expressing GFP-LacIR under the control of the inducible *GALS* promoter in the WT (wt ind., yAT2078), and in a *sir3∆* (*sir3∆ ind,.* yAT2370) strains. The data were normalized over *OGG1* (shown as mean ± s.e.m.; see Appendix A for the number of experiments). (**E**) Representative fluorescent images (maximal projection of a Z-stack) and their corresponding transmission images of strains bearing the *lys2::**lacO E-ADE2-I* locus and expressing GFP-LacIR under the constitutive *HIS3* promoter in the WT (wt const. yAT2059) and the *sir3∆* (*sir3∆ const.,* yAT3525) strains. Bar, 2 µm. (**F**) Cumulative distributions of the intensity of GFP foci in G1 cells, in strains bearing the *lys2::**lacO E-ADE2-I* locus and expressing GFP-LacIR under the constitutive *HIS3* promoter in the WT (wt const. yAT2059) and a *sir3∆* (*sir3∆ const.,* yAT3525) strains. *p* > 0.01 is non-significant (ns), *p* ≤ 0.00001(****) and see Appendix A for statistics.

## Data Availability

Not applicable.

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
