# Peer review of "An Inducible System for Silencing Establishment Reveals a Stepwise Mechanism in Which Anchoring at the Nuclear Periphery Precedes Heterochromatin Formation"

_cells, 2021, doi:10.3390/cells10112810_

Round 1

Reviewer 1 Report

Authors have solved all my concerns. The new text should be revised for typos. Otherwise, I am satisfied with the modifications done.

Author Response

We thank the reviewer for his positive assessment of our revised version. We have now corrected the typographical errors that we could find.

Reviewer 2 Report

In this resubmission, Loïodice et al. addressed my significant concerns, which revolved around the need for more biological replicates for presented data. These replicates provided statistical power behind their conclusions and strengthened the rigor of the paper. I thank the authors for performing them. The authors also clarified figures and text. “Data not shown” is now in a supplemental figure (S5D).

I appreciate that the authors attempted to perform the co-localization experiments of the array and Sir3 and understand the technical difficulties that hampered those efforts. While knowing whether or not the silenced array moves towards a Sir3 focus would have been interesting, it is not critical for publishing these findings.

The authors also fixed minor typos and phrasing.

I appreciate the authors pointing out where I was off target regarding the question of “Is this special about the LacO array?” I know it can be quite frustrating when reviewers provide criticism that the authors think was already addressed either in previous papers or in the current manuscript. I apologize for missing the experiment in Dubarry 2011 (lines 54-67 of response to reviewers) and misinterpreting normalization in Figure 3B/C (lines 103-107 of response to reviewers). Thank you for providing clarity in your response.

I recommend this study for publication in Cells.

Author Response

We thank the reviewer for his/her positive assessment of our revised version.

This manuscript is a resubmission of an earlier submission. The following is a list of the peer review reports and author responses from that submission.

Round 1

Reviewer 1 Report

In this work, Loïodice et al., have created a system able to induce ectopic silencing by introducing a lacI bound lacO array close to the ADE2 gene flanked by a HML silencer. Using this tool, authors have shown that the perinuclear anchoring precedes silencing of this locus by SIR complex, which requires several cell cycles to complete. In addition, they observe that the LacI binding to the DNA induce the phosphorylation of H2A and a nucleosome eviction process, which is counteracted by Sir3. The work presented here is of general interest and the experiments are well conducted. The text contains all the information required for a good understanding and is well written with a nice discussion.

However, I have just one major concern affecting the overall work. How can authors be sure that the endogenous lys2 locus without any modification has not a perinuclear localization after galactose induction? Maybe the different kinetic they observe for anchoring and silencing is because the initial perinuclear anchoring after galactose induction is happening as well to the endogenous locus. I may have missed some data, but the closer control I can see in this regard is the “constitutive” without silencer in Fig. 4B. However, I am not sure whether this control is done in glucose or galactose. Perhaps a good easy control would be to do the localization experiment with the constitutive strain without silencer shifted to galactose with different time points.  Different, and more direct, controls could be done to rule out this possibility. The most direct would be DNA FISH experiments to check the location of the lys2 locus upon galactose induction. A possible alternative could be to check what happen in a possible LacIR** allele.

Minor comments:

- ChIP experiments of Sir2 in the context of Figure 3 to check whether the delay on silencing is due to a delay on Sir2 recruitment would be interesting.

-Lines 272 and 293: authors mention that anchoring occurs at G1, while they show a cell cycle independent perinuclear anchoring. It is a bit confusing.

-Line 66: heterochromatin in budding yeast would be better as many yeasts has no Sir3 for example, and heterochromatin is more dependent on other factors.

-Line 80: reference.

-Line 175: final dot.

-Figure 3C. label αH4K16ac better?

-Is the legend of Fig. 5B wrong? In addition, I would suggest reviewing the colours and lines used in many graphs.

-Line 511: Figure S5G

Reviewer 2 Report

In this study, Loïodice et al., have moved a modified version of the HML locus to chr-II where it is not efficiently silenced as it would be at its native locus. Neat this locus ~2kb they have placed a 256x LacO array and can express a LacI-GFP fusion protein that binds this array either under a constitutive promoter or under an inducible promoter. When the Laci-GFP fusion is expressed the nearby modified HML locus becomes repressed. This locus also moves to the nuclear periphery. Because of the fine kinetic control of the inducible promoter, the authors are able to show that the move to the periphery surprisingly precedes silencing. The authors also show that phosphorylation of H2A precedes both silencing and peripheral localization. This is largely independent of the Sir3 silencing proteins. The reciprocal experiment showed that gH2A is not required for the silencing or peripheral localization of the locus. These two marks seem to help stabilize each other. The authors also show that during the early time points as H2A is being phosphorylated and some Sir3 is binding at the silencer the LacO array loses nucleosome occupancy. They hypothesize that this naked DNA, or large NDR, triggers stress and the beginning of the DNA repair process initiating the downstream cascade described above.

Overall, this is a well-thought-out and clever system and the authors rightfully acknowledge that this is an artificial system. Despite being artificial, the system allowed for nice temporal resolution of the steps in silencing and relocalizing a locus. I want to provide explicit praise for how clear and detailed the methods section is.

Major Concerns:

Looking at the number of experiments performed in the Supplemental Data some of the ChIP-qPCRS were clearly not performed with enough biological replicates. For example (Figure 3A) some induction time points with some primers were never performed or only performed once, but there appears to be a data point in the graph (It’s actually not clear). Many others were only performed with two biological replicates. It is not even clear how statistics for significance can be performed in these cases. I would like to see at least 3 biological replicates for all time points and conditions shown in the main figures at least. Figure 5B is another example. There is a single data point for Dmec1 ind at P4 45’. This needs to be repeated with three biological replicates. This should be also done for supplemental figures that conclusions are drawn from, but I am willing to allow a little more wiggle room there.

At the conclusion of the manuscript, I was left with one major question. Is this something special with LacI-GFP or is it universal to other arrays and systems? Will any array of tightly bound protein-DNA contacts result in nucleosome eviction and a similar phenomenon or is LacI-GFP interacting with something causing this all to happen? I can think of a couple of ways to address this: (1) Express LacI without the GFP and ask if silencing still occurs and if H2A is still phosphorylated. This will eliminate the GFP component of the fusion system. I understand that the relocalization experiments cannot be done under this system. (2) Insertion of another array such as a TetO array and recapitulate some of the findings with the LacO array. Neither of these requires all timepoints, really just the final long-term readout. A figure similar to Figure 1B and ChIP to gH2A will probably suffice.

The authors hypothesize that (line 670) “As mentioned above the perinuclear anchoring of the locus that occurs independently of SIR recruitment, could also contribute to SIR recruitment by bringing the locus in proximity to the pool of SIR proteins associated with silent loci enriched at the nuclear periphery.” I agree this is likely and there is an experiment the authors could do to strengthen this hypothesis. Track the locus over time and its association with Sir3 foci. This could most easily be done by crossing with a Sir3-GFP strain and crossing out the LacI-GFP with a LacI-CFP. If this hypothesis is correct then the locus should be in close proximity to a Sir3-GFP focus as Sir3 levels increase at the locus. Is the array only in proximity to a Sir3-GFP focus at 20 hours at maximal silencing?

The missing biological replicate issue must be addressed before publication. The other two major concerns would certainly strengthen the paper and relieve concerns that it is not completely an artifact of the LacO-LacI-GFP system and strengthen the model with fairly straightforward experiments that would not be too cumbersome.

Minor Concerns:

The manuscript needs some minor editing for typos and some English phrasing but overall is very well written. Eg. Line 351, 354

Line 288 “GFP spots were lower in G1 phase than in S phase” How do the authors distinguish S from G2 here? Should this be phrased as S/G2?

Figure 2D: Y-axis labeled ADE2 mRNA level over ACT1 but seems there is an additional normalization step not on the label (ADE2 of another strain?). Could this be made clearer in the figure?

Figure 3A “wt ind Raf” would “wt-non-ind” be clearer to those not intimately familiar with this induction system?

Figure 3B/C: It is not clear why one is normalized to the constitutive expression but the other is not. These should be consistent.

Around line 516 and elsewhere it is interesting in this system that Sir3 was not required for gH2A, binding when Kirkland and Kamakaka 2012 (PMID: 23733345), showed that at the native HMR locus the E silencer (and presumably then Sir proteins) was required for gH2A? Do the authors think this is a difference in systems or something else biological?

Line 603: The Rad53 data not shown should be shown in a supplemental figure.

Line 609-610 “However none of these factors is required for silencing and establishment in our system” Consistent with your findings is that they are also not required for silencing at HM despite being sufficient in ref 44 (Kirkland and Kamakaka 2012; PMID: 23733345).